# Consumption of over-the-Counter Drugs: Prevalence and Type of Drugs

**DOI:** 10.3390/ijerph18115530

**Published:** 2021-05-21

**Authors:** Eduardo Sánchez-Sánchez, Fátima Lucía Fernández-Cerezo, Jara Díaz-Jimenez, Manuel Rosety-Rodriguez, Antonio Jesús Díaz, Francisco Javier Ordonez, Miguel Ángel Rosety, Ignacio Rosety

**Affiliations:** 1Internal Medicine Department, Punta de Europa Hospital, Algeciras, 11207 Cádiz, Spain; 92fatima98@gmail.com; 2Instituto de Investigación e Innovación Biomédica de Cádiz (INiBICA), Hospital Universitario Puerta del Mar, Universidad de Cádiz, 11009 Cádiz, Spain; 3Doctoral School of the University of Cádiz (EDUCA), Campus Cádiz, Edificio Hospital Real (Primera Planta), Plaza Falla 8, 11003 Cádiz, Spain; luna_nueva17@hotmail.com; 4Medicine Department, School of Medicine, University of Cádiz, Plaza Fragela s/n, 11003 Cadiz, Spain; manuel.rosetyrodriguez@uca.es; 5Medicine Department, School of Nursing, University of Cadiz, Plaza Fragela s/n, 11003 Cadiz, Spain; antoniojesus.diaz@uca.es; 6Human Anatomy Department, School of Medicine, University of Cádiz, Plaza Fragela s/n, 11003 Cadiz, Spain; franciscojavier.ordonez@uca.es (F.J.O.); Ignacio.rosety@uca.es (I.R.); 7Move-It Research Group, Biomedical Research and Innovation Insitute of Cadiz, Puerta del Mar University Hospital, University of Cádiz, Plaza Fragela s/n, 11003 Cadiz, Spain; Miguelangel.rosety@uca.es

**Keywords:** nonprescription drugs, OTC drugs, OTC medication, over-the-counter medicines, self-care, self-medication

## Abstract

Recently, there has been an increase in the use of over-the-counter (OTC) drugs. The consumption of these medicines can be unsafe, as incorrect self-diagnosis or the ingestion of inappropriate doses can lead to side effects and the occurrence of adverse reactions and drug–drug interactions. A cross-sectional descriptive observational study was carried out, targeting the entire Spanish population by using an online questionnaire. The results showed that 78.9% of the subjects had previously taken or were currently taking OTC drugs. This consumption decreased as the age of the subjects increased, with a prevalence of 36.4% of subjects aged ≥ 71 taking OTC drugs. Analgesics were the most consumed OTC drugs (49.1%) especially in women, youngsters with non-formal educational qualifications, and individuals of a low–medium socioeconomic level residing in urban areas. Measures should be implemented to optimize the safe use of OTC drugs in order to avoid the occurrence of secondary events associated with the lack of knowledge related to their the usage.

## 1. Introduction

Recently, an increase has been observed in the use of pharmaceutical drugs by the general population [1]. However, this increase has not been associated with an improvement in health. According to the World Health Organization (WHO), more than 50% of the drugs consumed worldwide are prescribed or dispensed inappropriately and almost 50% of patients use these drugs incorrectly, resulting in increased morbidity and mortality [2,3].

Several factors have influenced this increase, namely, increased age, higher access to medication and increased self-diagnosis and treatment, which means the management of own health through all available resources, individual and/or collective [4]. This self-management of health encompasses the concept of self-medication, defined as the selection and use of drugs by individuals and/or caregivers to treat self-recognized or self-diagnosed conditions or symptoms. This self-medication can be an unsafe practice [5] due to inaccurate self-diagnosis or an inappropriate dosage intake that can result in side effects, adverse reactions, and pharmacological interactions [4]. The rate of self-medication has been reported to be up to 90% in developing countries. Improper self-medication increases the costs associated with health care.

Studies have shown that many factors (cultural, political, and environmental) influence the practice of self-medication for the prevention or treatment of symptoms and diseases [6]. The most common medical ailments for which self-medication is practiced include headache, joint pain, fever, cold and cough, allergy, heartburn and diarrhoea [7]. Increased access to medicines without improved health literacy increases the risk of misuse.

Among the drugs used most frequently by the population, there are some that require a prescription in order to be dispensed, as opposed to OTC drugs. Different countries have seen an increase in the consumption of OTC drugs, which can lead to their misuse [8]. In Spain, the purchase of prescription drugs has decreased and the consumption of OTC drugs has increased (Figure 1) [9,10]. In addition, periods of economic crisis lead to an increase in the number of patients taking OTC drugs [11].

Easy access to some medicines, such as OTCs, has caused the population to self-medicate, basing their decisions on previous prescriptions and on the advice of agents outside the health system, namely, family, friends, online social networks, internet search engines etc. [12,13]. In addition, mass media plays an important role, as it directly broadcasts advertisements concerning OTC drugs.

The aim of our study was to learn about the consumption of OTC drugs, the type of OTCs consumed, and the identification of the groups in the population that are at a higher risk of OTC drug consumption.

## 2. Materials and Methods

### 2.1. Selection of Participants and Study Design

We conducted a cross-sectional descriptive observational study that targeted the entire Spanish population. The population is composed of different demographic groups according to age, sex, and other demographic factors. It is important to note that the unemployment rate before the start of the study in Spain was 14.41%, influencing this rate on the socio-economic level and consequently on the acquisition of prescribed medicines (free of charge for unemployed) and OTC drugs.

To calculate the sample size, the data published by The National Statistical Institute (INE) on the 1st of July 2019 were taken as the reference, which reflected that the Spanish population comprised 47,100,936 inhabitants. Epidat 4.2 software (Galician Health Service, Galicia, Spain)) was used to calculate the sample size. The calculation of the sample size was carried out with a 95% confidence level and 3% precision, and since the expected proportion of change in the population was unknown, a 0.5 proportion was selected. The sample size was 203 subjects.

### 2.2. Instruments and Variables

The study variable selected was the consumption of OTCs, as an affirmative or negative answer. In addition, another study variable, type of OTCs consumed, was also recorded.

Several independent variables were selected, such as sociodemographic variables (gender, age, education level, and place of residence), economic variables (employment status and socioeconomic status), and health variables (presence of disease or illness and the consumption of medication).

In addition, information was collected concerning resources used to search for information on OTC drugs (television, internet, social networks, etc.) This last variable was collected by means of close-ended questions.

A self-administered questionnaire was developed to collect information about the different variables. The items included in the questionnaire are shown in the results tables.

### 2.3. Data Collection

Concerning the dissemination of the questionnaire, new communication technology using the Google questionnaire platform was employed, and social platforms on Twitter, Facebook, WhatsApp, and Instagram were used for the dissemination. The questionnaire was administered between March and April 2020. In this period, there were pandemic-related mobility restrictions in Spain (Lockdown COVID-19).

### 2.4. Statistical Analysis

The data obtained from the different variables were represented descriptively, by frequency and percentage. Subsequently, using a Chi-square test, we studied whether there were significant differences between the dependent variable and the independent variables, accepting a confidence level of 95%.

In addition, a multivariate logistic regression model was performed with all independent variables (gender, age, study level, employment status, socioeconomic status and place of residence) and OTC consumption. The statistical analysis was carried out with the R-Commander software version 2.7-1 (Hamilton, ON, Canadá).

## 3. Results

### 3.1. Participants and Sociodemographic and Economic Variables

A total of 728 responses to the questionnaire were obtained. Of these, one response was eliminated as it was incomplete, so the study was finally carried out using the remaining 727 responses.

Of the total sampled population under study, 75.1% were women (*n* = 546) and 24.9% were men (*n* = 181). The age group representing the highest percentage in our study were individuals aged 25 years (39.9%; *n* = 290), with ≥71 years of age being the group with the lowest number of members (1.5%; *n* = 11).

The number of participants who had completed their education or were currently studying at university level was 39.0% (*n* = 284), and only 3.1% (*n* = 23) did not possess any educational qualifications. In addition, 43.2% (*n* = 314) were working at the time of answering the questionnaire. A high percentage of subjects (60.8%) classified their economic level as low–medium. Most participants (74.5%, *n* = 542) resided among an urban population (Table 1).

### 3.2. Health Status and Medication Use

In total, 59.0% (*n* = 429) of the sample did not present any type of condition or disease, followed by 32.4% (*n* = 236) who reported a chronic condition (more than 6 months) and 8.5% (*n* = 62) who reported an acute condition (less than 6 months). In terms of gender, women had more acute and chronic conditions than men (9.0% vs. 7.2%, and 34.1% vs. 27.6%, respectively), although there were no statistically significant differences (*p* = 0.148). Chronic conditions were more prevalent in subjects with advanced age (71 years), who were retired, and those residing among a rural population. In addition, women consumed more medications than men (57.1% vs. 46.4%), which was mostly influenced by the following factors, subjects aged between 56 and 70 years (83.3%), no formal education (69.6%), and retirement (84.4%) (Table 2). Among the subjects taking medication, 61.6% (*n* = 244) were taking medication prescribed by their doctor on a regular basis, while 28.4% (*n* =152) were taking medication as needed, which had been previously prescribed or because they knew of its benefits. In the last 6 months, 56.5% (*n* = 411) of the sampled population reported taking medication sporadically, 35.7% (*n* = 260) took it regularly, and 7.7% (*n* = 56) did not take any medication. Sporadic use was more prevalent in men (63.0%), adults (26–40 years: 66.3%), with postgraduate education (68.9%), individuals with a low economic status (58.2%), students (66.9%), and those residing among an urban population (59.6%). Regular consumption of drugs was more prevalent in women (39.2%), aged 56–70 years (70.4%), with non-formal educational qualifications (56.5%), individuals with a high economic status (62.5%), retired individuals (71.9%), and those residing among a rural population (41.7%). Statistically significant differences between the different categories of each variable (*p* < 0.05) were confirmed.

### 3.3. OTC Consumption

There were differences in the age groups, with the consumption of OTC drugs decreasing with an increase in age. In total, 85.5% (*n* = 248) of subjects aged <25 years-old consume or have consumed OTC drugs, decreasing this percentage to 36.4% (*n* = 4) in subjects aged >71 years-old. Additionally, the subjects with graduate degrees consumed fewer OTC drugs (*n* = 70; 65.4%) when compared to those with postgraduate degrees (*n* = 39; 86.7%). Significant differences were also observed between different groups according to their employment status. Retired subjects indicated a lower consumption of OTC drugs (*n* = 19; 59.4%). Students were among the groups that indicated the highest consumption of OTC drugs (*n* = 222; 85.7%).

In addition, 72.6% (*n* = 528) responded that they had bought an OTC drug advertised on television (Table 3).

### 3.4. Type of OTC Drugs Consumed

Analgesics were the most consumed OTC drugs (*n* = 357; 49.1%), while antidepressants were the least commonly consumed OTC drugs (*n* = 3; 0.4%) (Figure 2). Although antidepressants are not considered OTC, they have been included in this analysis, as they are available from some online “pharmacies” and have sometimes also been purchased from conventional pharmacies [14].

In terms of gender, women consumed more analgesics and cough medicine than men (51.8% vs. 40.9%, and 16.7% vs. 14.4%, respectively). The rest of the drugs were consumed more by men, and this difference was greater in the “other” group, which includes all drugs not described above (difference of 7.9%). By age, young people (25 years of age) consumed more analgesics, with the consumption of antihistamines, gastroprotection, and “other” drugs increasing with age. Subjects without academic degrees consumed more analgesics (69.6%) than the rest of the population. The consumption of medicines for the common cold and antihistamines was higher in subjects with a higher level of education (postgraduates), at 24.4% and 8.9%, respectively. Employment status also indicated differences among the various groups, and the consumption of analgesics was more prevalent in students (55.6%). Socioeconomic status was also another factor that influenced consumption of OTC drugs. In this respect, significant associations were found between individual’s socioeconomic status and their consumption of drugs such as antibiotics, antidepressants, common cold drugs, and antihistamines (Table 4).

### 3.5. Self-Medication and Consumption of OTC Drugs

Questions related to the consumption of these medicines and self-medication were asked. When subjects were asked what action they took when they had a headache 66.2% (*n* = 481) responded that they self-medicated with an analgesic, 29.7% (*n* = 216) did not take anything and waited for it to go away on its own, and 4.1% checked with friends and family (1.1%; *n* = 8) or went to their doctor or pharmacist (3.0%; *n* = 22).

In addition, 14.7% (*n* = 107) responded that they took medication before experiencing any symptoms, using it as prevention before the onset of symptoms.

### 3.6. Search for Information.

A total of 58.5% (*n* = 425) of subjects surveyed, reported that they had searched the internet for information about medication. More specifically, 89.1% (*n* = 648) responded that when they had a headache, they searched for information to know the cause and/or the treatment. However, when they had doubts about a medication, 49.7% (*n* = 361) read the package leaflet, followed by 35.9% (*n* = 261) who went to a healthcare professional or pharmacist, 10.0% (*n* = 73) who searched the internet, and 4.4% (*n* = 32) who asked a family member and/or a friend. As there was a high percentage of patients who consulted the package leaflet, they were asked if they knew how to look for information in the leaflet and if they were aware of the warning symbols. In total, 77.3% (*n* = 562) of those surveyed knew how to look for information in the package leaflet.

### 3.7. OTC Drugs Odds Ratio and Independent Variables

After performing the multivariate regression model, the category “≥71 years” of the variable age, and the groups “non-formal education” and “university” in the variable level of studies presented statistical significances. Subjects aged ≥ 71 years are less likely to use OTC drugs (OR = 0.04; CI: 0.00–0.27, *p* < 0.001). However, the category “no education” and “university students” presented OR values of 4.23 (CI: 1.21–18.41, *p* = 0.033) and 1.84 (CI: 1.00–3.33, *p* = 0.045), respectively (Table 5).

## 4. Discussion

The results obtained show that the consumption of OTC drugs is high (78.9%), with higher consumption in subjects aged ≤ 25 years (85.5%), students (85.7%) and those with postgraduate degrees (86.7%). Of these medicines, analgesics are the most consumed medicines (49.1%).

OTC drugs are relatively safe, however inappropriate use can lead to complications and adverse events that affect public health [15]. Due to their greater availability and ease of acquisition, these drugs are consumed before or at the onset of symptoms. The percentage of consumption of OTC drugs obtained from our study results (78.9%) is higher than that reported in other countries such as Turkey (33%) and lower than Sweden (87%) [16], Czech Republic (92%), and Cyprus (97%) [17]. It is worth noting that almost six out of ten respondents had no medical condition or illness at all, and only 8.5% had an acute illness.

The present study found that several items (aged ≤ 25 years (85.8%); students (85.7%); postgraduate education (86.7%); medium-high economic status (82.8%); urban population (80.1%)) were associated with a higher consumption of OTC drugs. These results are similar to those obtained by other researchers, although in our study, differences with regard to gender were not obtained [18]. In contrast to a previous study conducted by Shah et al., subjects aged >65 years-old were not the major consumers of OTC drugs in Spain [19]. This difference may be due to this population having greater contact with prescribers of medicines and the fact that the medicines prescribed are free of charge. Subjects living among rural populations indicated a lower consumption of OTC drugs than those residing among urban populations. However, the consumption of drugs prescribed by medical practitioners was higher, as well as the presence of chronic diseases. These results, although at a lower percentage (23%), were similar to those obtained by Dutta et al. in a rural population [20]. This may be due to different factors such as: the availability of these drugs in pharmacies, the consumption of advertising through the media and social networks [21], the greater presence of acute illness, lifestyle, and living in an urban population.

Our results show that the groups with opposite levels of education, namely, subjects with no formal education qualifications and subjects pursuing university education were among those most likely to consume OTC drugs, with a probability of 4.23 and 1.84 times higher than the rest of the subjects, respectively. Conversely, in a previous study conducted in Eritrea, respondents with elementary and non-formal education were fifteen times more likely to consume OTC drugs compared to those with a higher education [15].

Pain relief is a prominent factor influencing the use of these drugs internationally, with headaches, low back pain, and joint pain being the triggers leading to the use of these drugs. Analgesics are the most frequently consumed OTC drugs [22]. Therefore, analgesics constitute the largest OTC market worldwide [23]. The consumption of NSAIDs in our study population was lower than that reported in other studies, such as the study conducted by Arain et al. in 2019, where they concluded that 68.9% of the population took OTC NSAIDs. In addition, analgesics can lead to side effects that affect the health of the consumer. In the aforementioned study, one third of the population was unaware of contraindications associated with analgesics [24]. After analgesics, antitussives and common cold drugs were most commonly consumed by our subjects, which is different to the results published by other authors, who reported that analgesics are the most consumed, followed by antipyretics and antibiotics [25]. In addition, we recall again that antidepressants are not considered as OTC medicines but are present in this study because of their possible acquisition without prescription in some cases. Hence, the number of subjects who have taken this medicine without a prescription is very low (*n* = 3).

Another important aspect is the presentation of OTC drugs and the information contained in the product. In this respect, the package leaflet played a fundamental role, as it provided the user with essential information about the drug. A total of 49.7% of the participants in our study reported that they read the package leaflet when they had doubts about a drug, but only 77.3% knew how to properly look for information in the leaflet and understand the warning symbols. These results were different from those obtained in previous studies, which concluded that 59.6% read the package leaflets regularly, although more than half of the leaflets (57.4%) had legibility problems [26].

Direct-consumer advertising by the pharmaceutical industry is important, especially in relation to OTC drugs. This type of advertising is done through advertisements on TV, radio, and other sources [27]. Our results showed that 27.4% had consumed an OTC drug advertised on TV. In addition, many consumers use the internet to search for information concerning the use of OTC drugs or products. A total of 58.5% of the respondents reported that they used the internet to search for information about drugs, reflecting the importance of e-health literacy [28].

The low representativeness of the older population (≥56 years) and of male subjects are two of the study’s limitations. This may be due to the use of an online questionnaire, as opposed to a face-to-face interview, but it was a strategy used to overcome restrictions and avoid contact during the COVID-19 pandemic. The use of online questionnaires as a method of survey collection during the pandemic has increased exponentially. This study did not ask about the frequency or use of different medications as this was not the main objective of the study (i.e., this study aimed to focus on the consumption and type of OTC drugs and at-risk population consuming these medicines) but this may be seen as a study limitation.

## 5. Conclusions

The consumption of OTC drugs in the study population is very high, much higher than the prevalence of acute or chronic diseases and the consumption of prescription drugs in this population. Subjects with a non-formal education or a university education are the population groups that are most likely to consume OTC drugs.

Among OTC drugs, analgesics are the most consumed, followed by antitussives, and drugs for the common cold.

National health policies concerning the use of OTC drugs should be targeted at these populations and drug groups in order to optimize their use and avoid the occurrence of side effects associated with the lack of knowledge in the use of these drugs. In addition, these policies should improve e-health literacy, considering the increase in the number of people who use the internet or social networks as health resources to search for information.

Further research in this area should be carried out, covering a sample of the older population.

## Figures and Tables

**Figure 1 ijerph-18-05530-f001:**
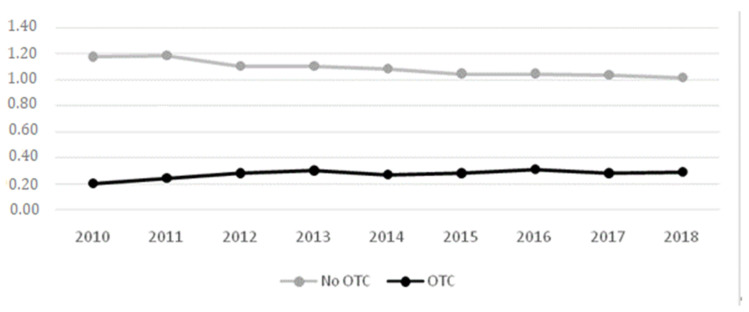
Pharmaceutical expenditure, percentage over the Growth Domestic Product (GDP). Data obtained from Sistema de Cuentas del Ministerio de Sanidad, Consumo y Bienestar Social (2003–2018) [10].

**Figure 2 ijerph-18-05530-f002:**
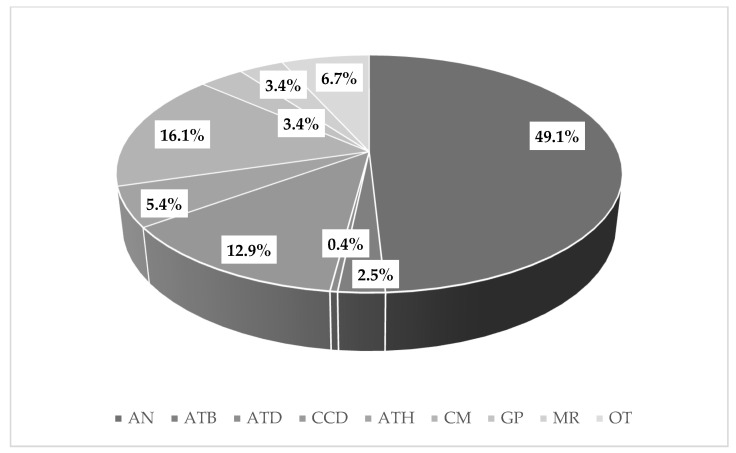
Major types of OTC drugs. AN: Analgesics; ATB: Antibiotics; ATD: Antidepressants; CCD: Common cold drugs; ATH: Antihistamines; CM: Cough medication; GP: Gastroprotection; MR: Muscle relaxant; OT: Others.

**Table 1 ijerph-18-05530-t001:** Sociodemographic and economic variables of the total sampled population.

Variables	*N*	%
**Gender:**		
• Female	546	75.1
• Male	181	24.9
**Age:**		
• ≤25 years-old	290	39.9
• 26–40 years-old	169	23.3
• 41–55 years-old	203	27.9
• 56–70 years-old	54	7.4
• ≥71 years-old	11	1.5
**Study level:**		
• No formal education	23	3.1
• Primary education	107	14.7
• Secondary education	50	6.9
• Vocational training	128	17.6
• Upper secondary education	90	12.4
• University degree	284	39.0
• Postgraduate degree	45	6.2
**Employment status:**		
• Student	266	36.6
• Unemployed	115	15.8
• Employed	314	43.2
• Retired	32	4.4
**Socioeconomic status:**		
• Low	79	10.9
• Low–medium	442	60.8
• Medium–high	198	27.2
• High	8	1.1
**Place of residence:**		
• Rural	185	25.4
• Urban	542	74.5

Note: Results expressed as number of respondents (*n*) and percentage (%).

**Table 2 ijerph-18-05530-t002:** Health status and medication use of the population sample.

	Do You Suffer from Any Condition?				Are You on Any Medication?		
	NO	Acute	Chronic	*p*-Value	YES	NO	*p*-Value
**Gender:**				0.148			0.011 *
• Female	311 (57.0)	49 (9.0)	186 (34.1)		312 (57.1)	234 (42.9)	
• Male	118 (65.2)	13 (7.2)	50 (27.6)		84 (46.4)	97 (53.6)	
**Age:**				<0.001 ***			<0.001 ***
• ≤25 years-old	191 (65.9)	28 (9.7)	71 (24.5)		123 (42.4)	167 (57.6)	
• 26–40 years-old	113 (66.9)	17 (10.1)	39 (23.1)		78 (46.2)	91 (53.8)	
• 41–55 years-old	103 (50.7)	13 (6.4)	87 (42.9)		141 (69.5)	62 (30.5)	
• 56–70 years-old	20 (37.0)	3 (5.6)	31 (57.4)		45 (83.3)	9 (16.7)	
• ≥71 years-old	2 (18.2)	1 (9.1)	8 (72.7)		9 (81.8)	2 (18.2)	
**Study level:**				0.258			0.001 **
• Non-formal education	11 (47.8)	3 (13.0)	9 (39.1)		16 (69.6)	7 (30.4)	
• Primary education	60 (56.1)	7 (6.5)	40 (37.4)		73 (68.2)	34 (31.8)	
• Secondary education	28 (56.0)	8 (26.0)	14 (18.0)		32 (64.0)	18 (36.0)	
• Vocational training	81 (63.3)	12 (9.4)	35 (27.3)		72 (56.2)	56 (43.8)	
• Upper secondary education	44 (48.9)	9 (10.0)	37 (41.1)		47 (52.2)	43 (47.8)	
• University degree	180 (63.4)	9 (6.7)	85 (29.9)		129 (45.4)	155 (54.6)	
• Postgraduate degree	25 (55.6)	4 (8.9)	16 (35.6)		27 (60.0)	18 (40.0)	
**Employment status:**				<0.001 ***			<0.001 ***
• Student	178 (66.9)	24 (9.0)	64 (24.1)		112 (42.1)	154 (57.9)	
• Unemployed	54 (47.0)	14 (12.2)	47 (40.9)		80 (69.6)	35 (30.4)	
• Employed	186 (59.2)	23 (7.3)	105 (33.4)		177 (56.4)	137 (43.6)	
• Retired	11 (34.4)	1 (3.1)	20 (62.5)		27 (84.4)	5 (15.6)	
**Socioeconomic status:**				0.408			0.329
• Low	47 (59.5)	6 (7.6)	26 (32.9)		43 (54.6)	36 (45.6)	
• Low–medium	255 (57.7)	43 (9.7)	144 (32.6)		248 (56.1)	194 (43.9)	
• Medium–high	123 (62.1)	11 (5.6)	64 (32.3)		99 (50.0)	99 (50.0)	
• High	4 (50.0)	2 (25.0)	2 (25.0)		6 (75.0)	2 (25.0)	
**Place of residence:**				0.004 **			0.080
• Rural	93 (50.3)	14 (7.6)	78 (42.2)		111 (60.0)	74 (40.0)	
• Urban	336 (62.0)	48 (8.9)	158 (29.2)		285 (52.6)	257 (47.4)	

Note: Results expressed as number of respondents (percentage). Statistical significance * 0.05, ** 0.01; *** 0.001.

**Table 3 ijerph-18-05530-t003:** Prevalence of OTC drugs use.

Variables	YES (*n*, %)	NO (*n*, %)	*p*-Value
**Gender:**			0.984
• Female	431 (78.9)	115 (21.1)	
• Male	143 (79.0)	38 (21.0)	
**Age:**			<0.001 ***
• ≤25 years-old	248 (85.5)	42 (14.5)	
• 26–40 years-old	140 (82.8)	29 (17.2)	
• 41–55 years-old	148 (72.9)	55 (27.1)	
• 56–70 years-old	34 (63.0)	20 (37.0)	
• ≥71 years-old	4 (36.4)	7 (63.6)	
**Study level:**			0.001 **
• Non-formal education	19 (82.6)	4 (17.4)	
• Primary education	70 (65.4)	37 (34.6)	
• Secondary education	36 (72.0)	14 (28.0)	
• Vocational training	104 (81.2)	24 (18.2)	
• Upper secondary education	66 (73.3)	24 (26.7)	
• University degree	240 (84.5)	44 (15.5)	
• Postgraduate degree	39 (86.7)	6 (13.3)	
**Employment status:**			<0.001 ***
• Student	228 (85.7)	38 (14.3)	
• Unemployed	83 (72.2)	32 (27.8)	
• Employed	244 (77.7)	70 (22.3)	
• Retired	19 (59.4)	3 (40.6)	
**Socioeconomic status:**			0.468
• Low	62 (78.5)	17 (21.5)	
• Low–medium	342 (77.4)	100 (22.6)	
• Medium–high	164 (82.8)	34 (17.2)	
• High	6 (75.0)	2 (25.0)	
**Place of residence:**			0.205
• Rural	140 (75.7)	45 (24.3)	
• Urban	434 (80.1)	108 (19.9)	

**Note:** Results expressed as number of respondents (percentage). Statistical significance ** 0.01; *** 0.001.

**Table 4 ijerph-18-05530-t004:** Major types of OTC drugs.

Variables	OTC Drugs (%)									
	AN	ATB	ATD	CCD	ATH	CM	GP	MR	OT	*p*
**Gender:**										0.008 **
• Female	51.8	2.4	0.4	12.6	5.5	16.7	3.1	2.7	4.8	
• Male	40.9	2.8	0.6	13.8	5.0	14.4	4.4	5.5	12.7	
**Age:**										<0.001 ***
• ≤25 years-old	53.4	4.1	0.0	6.6	5.2	17.6	2.4	2.4	8.3	
• 26–40 years-old	49.7	3.6	0.0	18.9	4.7	11.2	2.4	5.3	4.1	
• 41–55 years-old	44.3	0.0	1.0	18.2	6.4	15.8	4.4	4.4	5.4	
• 56–70 years-old	42.6	0.0	1.9	9.3	3.7	27.8	7.4	0.0	7.4	
• ≥71 years-old	45.5	0.0	0.0	9.1	9.1	0.0	9.1	0.0	27.3	
**Study level:**										0.006 **
• Non-formal education	69.6	0.0	0.0	4.3	4.3	13.0	0.0	4.3	4.3	
• Primary education	38.3	1.9	1.9	14.0	5.6	21.5	4.7	5.6	6.5	
• Secondary education	38.0	6.0	0.0	18.0	6.0	18.0	2.0	6.0	6.0	
• Vocational training	39.1	1.6	0.0	18.0	5.5	24.2	3.1	3.9	4.7	
• Upper secondary education	53.3	3.3	0.0	10.0	6.7	14.4	6.7	0.0	5.6	
• University degree	58.1	1.8	0.0	9.2	4.2	12.7	2.8	2.8	8.5	
• Postgraduate degree	40.0	6.7	2.2	24.4	8.9	4.4	4.4	4.4	6.7	
**Employment status:**										<0.001 ***
• Student	55.6	4.5	0.0	6.8	4.5	14.3	3.0	2.6	8.6	
• Unemployed	45.2	1.7	0.9	13.0	3.5	27.0	5.2	2.6	0.9	
• Employed	45.9	1.3	0.3	18.2	7.3	13.1	2.9	4.8	6.4	
• Retired	40.6	0.0	3.1	12.5	0.0	21.9	0.2	0.0	15.6	
**Socioeconomic status:**										<0.001 ***
• Low	43.0	7.6	00	15.2	3.8	19.0	1.3	3.8	6.3	
• Low–medium	50.2	1.4	0.5	11.5	4.5	16.7	4.3	3.8	7.0	
• Medium–high	49.0	2.0	0.0	15.7	8.1	13.6	2.5	2.5	6.6	
• High	50.0	25.0	12.5	0.0	0.0	12.5	0.0	0.0	0.0	
**Place of residence:**										0.574
• Rural	45.4	2.7	0.0	11.9	8.1	17.8	2.7	3.8	7.6	
• Urban	50.4	2.4	0.6	13.3	4.4	15.5	3.7	3.3	6.5	

AN: Analgesics; ATB: Antibiotics; ATD: Antidepressants; CCD: Common cold drugs; ATH: Antihistamines; CM: Cough medicine; GP: Gastroprotection; MR: Muscle relaxants; OT: Others. Statistical significance ** 0.001. *** 0.001.

**Table 5 ijerph-18-05530-t005:** Odds Ratio Values.

Variables	OR	CI 95%	*p*
**Gender (Male)**			
• Female	1.05	0.66–1.63	0.830
**Age (≤25 years-old)**			
• 26–40 years-old	0.80	0.38–1.64	0.550
• 41–55 years-old	0.59	0.26–1.25	0.179
• 56–70 years-old	0.26	0.09–0.68	0.006
• ≥71 years-old	0.04	0.00–0.27	<0.001 ***
**Study level (Upper Secondary education)**			
• Non-formal education	4.23	1.21–18.41	0.033 *
• Primary education	0.58	0.49–1.94	0.963
• Secondary education	0.96	0.43–2.21	0.932
• Vocational training	1.62	0.82–3.10	0.162
• University degree	1.84	1.00–3.33	0.045 *
• Postgraduate degree	2.59	0.95–8.06	0.076
**Employment Status (Unemployed)**			
• Student	1.20	0.52–2.67	0.654
• Employed	1.17	0.68–1.98	0.563
• Retired	1.82	0.58–6.12	0.311
**Socioeconomic status (High)**			
• Low	1.12	0.13–6.58	0.899
• Low–medium	1.01	0.12–5.55	0.985
• Medium–high	1.34	0.15–7-73	0.753
**Place of residence (Rural)**			
• Urban	1.79	0.76–1.79	0.448

**OR: Odds Ratio; CI: Confidence Interval.** Note: Statistical significance * 0.05, *** 0.001.

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
