# Peer review of "Consumption of over-the-Counter Drugs: Prevalence and Type of Drugs"

_ijerph, 2021, doi:10.3390/ijerph18115530_

Round 1

Reviewer 1 Report

This manuscript provides us information about the consumption of drugs whose dispensing or administration does not require a medical prescription in the Spanish population. This type of studies can be an important contribution to the health system. However, I would like to make a few suggestions for revision”:

OTC is a product, drug, or medicinal special specialty whose dispensing or administration does not require a medical prescription, and according to my knowledge the WHO Anatomical therapeutic chemical classification classified them in the following categories: analgesics, laxatives, antithrombotic agents, antacids, cough and cold preparations, antihistamines, dermatological, throat preparations, nasal preparations, and antidiarrheals

I understand that the dispensing of antidepressants without a prescription is a current practice in some countries. However, in my point of view antidepressants, s cannot be considered OTC products.

I suggest a clarification of what is OTC drugs and a clear justification for the inclusion of antidepressant in the study.

  1. Considering that almost:

20% of the Spanish population are over 65

15% of the Spanish population are under 15 years

We can affirm that near 35% of the population had limited access to the study due to the methodology chosen which in turn have been influenced by the pandemic context that we are living in. For these reasons, the authors cannot generalize the conclusions to all Spanish populations.

Moreover, I believed that this study has relevant information that can be used to help to design National health policies.

Author Response

Dear Reviewer,
thank you again for taking the time to review our manuscript and for your comments.
The points you requested have been modified or added, improving the comprehensibility of the manuscript.
We hope that these modifications will resolve the doubts raised in the previous review. 
Thank you very much for your time.
Kind regards.

Reviewer 2 Report

Dear authors,
thank you very much for your good predisposition in addressing the comments and suggestions. I only have four minor suggestions to add:
1. In your response to the comment #2 you said that "increased access to medicines without improving health literacy increases..." I agree, and I suggest that you include your own comment in the manuscript (not only in your response).
2. With regard to the sample size calculation, I suggest to include in the text the name and software version (Epidat 4.2), particularly because it was developed by public institutions and aimed at epidemiologists and other health professionals (within the framework of PAHO).
3. Despite that you said that the lack of representation of one sex over the other was added as a limitation in the discussion, I cannot found such a comment in the text. If missing, please included.
4. Lastly, I agree that antidepressants are sometimes acquired by patients without a proper prescription, but that does not mean that they are OTC medications. Please change that particular sentence, or add a clarification similar to the one in your response.

Author Response

Dear Reviewer,
thank you again for taking the time to review our manuscript and for your comments.
The points you requested have been modified or added, improving the comprehensibility of the manuscript.
We hope that these modifications will resolve the doubts raised in the previous review. 
Thank you very much for your time.
Kind regards.

This manuscript is a resubmission of an earlier submission. The following is a list of the peer review reports and author responses from that submission.

Round 1

Reviewer 1 Report

Consumption of over-the-counter drugs in the Spanish population: prevalence and type of drugs.

Use of over the counter (OTC) drugs is an important public health concern, indicating possibly unmet health needs by the medical system or medication overuse and possibly leading to serious adverse effects of medications or medication interaction. The authors thus are to be commended for tackling this concern in Spain.

I do have some problems with the presentation of the study and its results. I would have really liked to be able to access the main questions of the online questionnaire. For instance, for which time period was use of OTC medication to be indicated? Was there a question on the frequency or quantity of use? The result that about 79% of the participants indicated OTC medication use might also hint at the fact that occasional use over a long period was included, e.g. taking an analgesic several times a year. More precision in this respect would allow us to better appreciate the results.

The authors performed logistic regression to see whether or which participant characteristics were associated with the fact of having used OTC medication. In my, t, opinion, the authors do not present correctly the association with participants’ age: participants aver 71 years had a strong and significantly reduced risk of using OTC medications (OR=0.04, p<0.001): this is important – and reassuring – information, possibly related to more frequent contacts with prescribers and gratuity of prescription medication at that age. The association of OTC medication use with education is U shaped and the risk is considerably higher for persons with few years of education (the term “no education” should be avoided) but also for those with many years of education, possibly for different reasons: some literature-based reflection on this would be appropriate in the discussion.

Finally, figure 1 in the introduction shows the changes in expenditures for OTC and prescription medications in Spain between 2003 and 2017: these numbers on their own do not give us information on the numbers of  users, however.

Author Response

Dear reviewer,

Firstly, we appreciate the time dedicated to our manuscript, as well as the clarifications you request, which help us to understand the doubts that a future reader may have, if the manuscript gets published.

Secondly, we answer to the questions that you have made, with aim of resolving doubts raised by our manuscript.

I do have some problems with the presentation of the study and its results. I would have really liked to be able to access the main questions of the online questionnaire.

In order to answer your questions, we send you the link to the questionnaire used to carry out the study: https://docs.google.com/forms/d/e/1FAIpQLSd2nQZN_n1DVPiviRBaQCi-5MvzuAFVRlBLCpongChIviq-AA/viewform?usp=sf_link.

We did not add the items of the questionnaire to the text, because we thought that by appearing in the different tables, it was easier to read.

For instance, for which time period was use of OTC medication to be indicated? Was there a question on the frequency or quantity of use?

No time period was indicated, only whether they had used any over-the-counter or prescription medicines. No questions were asked about the frequency or quantity of use, as the aim of this study was to analyses whether these medicines were consumed, what types of medicines and to find out the age groups with the highest risk of consumption.

In addition to the limitations, no variables related to the frequency or amount of OTC drugs use were collected. Perhaps, other researchers reading the manuscript will consider research in this field and correct this limitation. As this is an anonymous study, we cannot contact respondents to add these questions, but it is something that should be considered, as you say, in order to know exactly the frequency and use of these drugs.

The result that about 79% of the participants indicated OTC medication use might also hint at the fact that occasional use over a long period was included, e.g. taking an analgesic several times a year. More precision in this respect would allow us to better appreciate the results.

Your comments have been added in order to make the results more visible.

The authors performed logistic regression to see whether or which participant characteristics were associated with the fact of having used OTC medication. In my, t, opinion, the authors do not present correctly the association with participants’ age: participants aver 71 years had a strong and significantly reduced risk of using OTC medications (OR=0.04, p<0.001): this is important – and reassuring – information, possibly related to more frequent contacts with prescribers and gratuity of prescription medication at that age. The association of OTC medication use with education is U shaped and the risk is considerably higher for persons with few years of education (the term “no education” should be avoided) but also for those with many years of education, possibly for different reasons: some literature-based reflection on this would be appropriate in the discussion.

This point has been modified in the results section in order to improve its comprehensibility. Suggested modifications have been made in the discussion, which have improved the manuscript.

The term "no education" has been changed to "non-formal education".

Finally, figure 1 in the introduction shows the changes in expenditures for OTC and prescription medications in Spain between 2003 and 2017: these numbers on their own do not give us information on the numbers of users, however.

Figure 1 has been modified and updated by adding expenditure over GDP, which better reflects pharmaceutical expenditure.

Once again, we appreciate the time and attention dedicated to our manuscript. We really hope we have reached your expectations, with the modifications made and that the explanations to those that we have not modified be considered as appropriate.

Kind regards.

Reviewer 2 Report

The manuscript is well written, and addresses an interesting topic, where updates on numbers and statistics are always welcome by the readers. 

However, there are great biases in the sample (see comments on the pdf file) that jeopardize the soundness of the conclusions. Analogously, the title is too ambitious since Spanish population is not well represented by the sample.

Therefore, I think that the authors should carefully revise two main aspects:

  • The statistical analysis (their interpretation and conclusions)
  • The drug classification, clearly discriminating between "real" OTC drugs and self-administered prescription drugs.

Lastly, I suggest to include more graphs in the manuscript, as they are very powerful tools to interpret, easily and quickly, descriptive statistics that are cumbersome to read in a table or in a text.

Author Response

Dear reviewer,

Firstly, we appreciate the time dedicated to our manuscript, as well as the clarifications you request, which help us to understand the doubts that a future reader may have, if the manuscript gets published.

Secondly, we answer to the questions that you have made, with aim of resolving doubts raised by our manuscript.

As you suggested, we have made some changes in order to improve the wording and validity of the manuscript. We attach the pdf you sent us with the answers to your clarifications, thanking you for the time you have devoted to it.

So far, he has been the most thorough reviewer we have met, and that helps us to improve the manuscript and to improve ourselves for future manuscripts. We have found him to be a very thorough review with pertinent clarifications.

Once again, we appreciate the time and attention dedicated to our manuscript. We really hope we have reached your expectations, with the modifications made and that the explanations to those that we have not modified be considered as appropriate.

Kind regards.

Reviewer 3 Report

Thank you for the opportunity to review this manuscript!  The study topic about the consumption of OTC is very relevant and useful to develop strategies to promote patient safety

This manuscript provides us information about the consumption of OTC products and gives us « a little perspective about the reality in Spain. The data provided in this study can help to design strategies to promote the safe use of OTC products and consequently improve patients' safety.  However, the article needs to be worked on. I would like to make a few suggestions for revision”:

  1. The authors “conducted a cross-sectional descriptive observational study that targeted the entire Spanish population”, and collected the data using new communication technologies. Considering that Spain is a large country with seventeen Autonomous communities, how the authors ensured that the applied questionnaires have been fulfilled by participants from the different autonomous communities to conclude that the data are representative of the Spanish population

  1. The J. Environ. Res. Public Health is an international journal and this work will be disseminated across the world characteristics of research areas should be written in detail for readers in the world.

Discussion

  1. In the first paragraph of the discussion the authors mentioned: “Subjects who were of advanced age had a higher prevalence of chronic diseases and consumption of medications prescribed by their doctor for the management of these diseases”.

Considering that aging is associated with the loss of resilience and the increase of comorbidities that prompt the consumption of medicines this data was not surprising.

I suggest that in the first paragraph of the discussion the most relevant data be emphasized.

I suggest that the discussion be written following the checklist from cross-sectional studies. https://www.strobestatement.org/fileadmin/Strobe/uploads/checklists/STROBE_checklist_v4_cross-sectional.pdf

4- line 219-220 authors report “Due to their greater availability and ease  of acquisition, these drugs are consumed as soon as any symptom or sign of acute illness  appears”.

How the authors obtained this information?

  1. Line 228 “urban population 228 (80.1%)) were associated to higher consumption of OTC drugs.

Can you explain based on your data and in the literature why urban populations are associated with higher consumption of OTC drugs?

Author Response

Dear reviewer,

Firstly, we appreciate the time dedicated to our manuscript, as well as the clarifications you request, which help us to understand the doubts that a future reader may have, if the manuscript gets published.

Secondly, we answer to the questions that you have made, with aim of resolving doubts raised by our manuscript.

This manuscript provides us information about the consumption of OTC products and gives us « a little perspective about the reality in Spain. The data provided in this study can help to design strategies to promote the safe use of OTC products and consequently improve patients' safety.  However, the article needs to be worked on. I would like to make a few suggestions for revision”:

  1. The authors “conducted a cross-sectional descriptive observational study that targeted the entire Spanish population”, and collected the data using new communication technologies. Considering that Spain is a large country with seventeen Autonomous communities, how the authors ensured that the applied questionnaires have been fulfilled by participants from the different autonomous communities to conclude that the data are representative of the Spanish population

The questionnaire was disseminated at national level, and the variable "autonomous community" was collected, but as it was not a compulsory question, some respondents did not answer and it was subsequently eliminated due to the high non-response rate. Other variables such as "study level", "employment status", "socioeconomic status", "place of residence", which can define the different population that is part of Spain, were collected.

  1. The J. Environ. Res. Public Health is an international journal and this work will be disseminated across the world characteristics of research areas should be written in detail for readers in the world.

As he comments, the IJERPH is an international journal and therefore, variables such as "study level", "employment status", "socioeconomic status", "place of residence" have been defined, which help the reader to better understand the study population. In the section, 2.1. Selection of Participants and Study Design a new text has been added. In addition, in the introduction to figure 1, we can see how the Spanish population behaves with respect to the purchase of OTC drugs.

Discussion

  1. In the first paragraph of the discussion the authors mentioned: “Subjects who were of advanced age had a higher prevalence of chronic diseases and consumption of medications prescribed by their doctor for the management of these diseases”.

Considering that aging is associated with the loss of resilience and the increase of comorbidities that prompt the consumption of medicines this data was not surprising.

I suggest that in the first paragraph of the discussion the most relevant data be emphasized.

I suggest that the discussion be written following the checklist from cross-sectional studies. https://www.strobestatement.org/fileadmin/Strobe/uploads/checklists/STROBE_checklist_v4_cross-sectional.pdf

Following your suggestion, the beginning of the discussion has been modified. In addition, the entire discussion has been revised and the checklist of cross-sectional studies has been followed as recommended by you.

4- line 219-220 authors report “Due to their greater availability and ease  of acquisition, these drugs are consumed as soon as any symptom or sign of acute illness  appears”.

How the authors obtained this information?

This sentence has been modified because it did not reflect the results obtained. In section 3.5 Self medication and consumption OTC (added after reviewers' comments) we can see how the study sample took OTC medicines even before presenting with a sign or at the beginning of the sign.

  1. Line 228 “urban population 228 (80.1%)) were associated to higher consumption of OTC drugs.

Can you explain based on your data and in the literature why urban populations are associated with higher consumption of OTC drugs?

Possible factors influencing increased consumption of OTC drugs in the urban population have been added in the text.

Once again, we appreciate the time and attention dedicated to our manuscript. We really hope we have reached your expectations, with the modifications made and that the explanations to those that we have not modified be considered as appropriate.

Kind regards.